# Quantum lock-in force sensing using optical clock Doppler velocimetry

Ravid Shaniv[1] & Roee Ozeri[1]

Force sensors are at the heart of different technologies such as atomic force microscopy or inertial sensing. These sensors often rely on the measurement of the displacement amplitude of mechanical oscillators under applied force. The best sensitivity is typically achieved when the force is alternating at the mechanical resonance frequency of the oscillator, thus increasing its response by the mechanical quality factor. The measurement of low-frequency forces, that are below resonance, is a more difficult task as the resulting oscillation amplitudes are significantly lower. Here we use a single-trapped $^{88}Sr^+$ ion as a force sensor. The ion is electrically driven at a frequency much lower than the trap resonance frequency. We measure small amplitude of motion by measuring the periodic Doppler shift of an atomic optical clock transition, enhanced using the quantum lock-in technique. We report frequency force detection sensitivity as low as $2.8 \times 10^{-20}\,NHz^{-1/2}$.

[1] Department of Physics of Complex Systems, Weizmann Institute of Science, Rehovot 7610001, Israel. Correspondence and requests for materials should be addressed to R.S. (email: ravid.shaniv@weizmann.ac.il) or to R.O. (email: roee.ozeri@weizmann.ac.il).

Force sensors are important for many applications ranging from computer game consoles to precision measurements of fundamental physics. Most force sensors rely on the displacement of mechanical oscillators as an estimator for the applied force. Trapped atomic ions can be well approximated as harmonic mechanical oscillators that are highly sensitive to electrical forces. Moreover, due to the extremely high-quality factor of ion traps, even very small forces that are resonant with the ion-trap harmonic frequency can yield relatively large displacements. The ion displacement amplitude is usually measured through its associated Doppler shift on an optical transition[1]. Previously, ion-trap experiments have demonstrated yocto-Newton range measurement sensitivity[2,3] of forces that were alternating at frequencies that varied between 50 and 900 kHz. Measuring forces that oscillate at frequencies much lower than this range is a challenge if one wishes to maintain the mechanical resonance condition. This is because as the trap weakens and the resonance frequency is lowered, the ion departs from the Lamb-Dicke regime that is a pre-condition to many trapped-ions control techniques. Moreover, low trap frequencies result-in faster heating times and a loss of sensitivity to displacements. If one wishes to measure forces that oscillate at a frequency of a few hundreds of Hz, for example, there is no alternative but to measure this force when it is off-resonance with mechanical resonance at a much higher frequency, resulting in significantly smaller oscillation amplitude. The measurement of low-frequency forces will therefore necessitate high-sensitivity Doppler-shift measurement techniques.

The most accurate frequency measurements to date have been performed on narrow optical clock transitions in laser-cooled atoms or ions. The state-of-the-art frequency comparisons are with fractional uncertainty in the $10^{-18}$ range[4-6]. An optical clock transition in a single-trapped ion can therefore serve as an excellent means to measure small Doppler shifts. Previously, periodic Doppler shifts on optical clock transitions, resulting in motional sidebands at the trap harmonic frequency, have been used for precision ion thermometry and sideband cooling. Similarly, motional sidebands can also be used for ion micromotion detection and compensation. The detection of oscillatory motion through its associated periodic phase modulation is ultimately limited by the transition phase noise at the relevant frequency. Since optical atomic clocks with transition linewidths narrower than 1 Hz have been demonstrated, forces that are oscillating in the 10–1,000 Hz range can be readily detected.

Sideband spectroscopy is a very useful measurement technique. However, when the phase modulation amplitude is small the sideband decreases linearly with the modulation depth. Moreover, the small sideband is often difficult to differentiate from the noisy background. An efficient method of measuring an oscillating signal by a quantum probe, in the presence of noise, is the Quantum lock-in technique[7]. Here, a probe superposition is time modulated by a Hamiltonian term, for example, $\hat{\sigma}_x$, that does not commute with the noise and signal operators, for example, $\hat{\sigma}_z$. The resulting dynamics decouples the probe phase evolution from any frequency component that does not spectrally overlap with the probe modulation and enables a significant prolongation of the probe coherence. On the other hand, frequency components that match some part of the modulation spectrum yield a phase that accumulates as the probe evolves with time. If the desired signal spectrally overlaps with the probe modulation, it imprints a coherently accumulated phase signal that can be subsequently measured. Quantum lock-in detection of different probes has recently led to high-precision magnetometery, as well as electric field

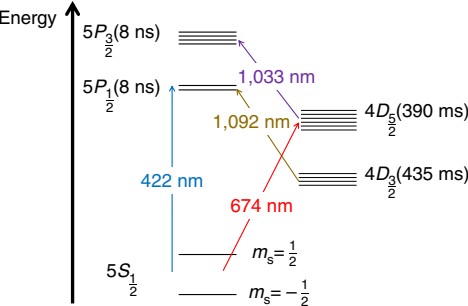

**Figure 1 | Schematic diagram of atomic levels.** $^{88}Sr^+$ ion relevant level scheme, levels lifetime and laser wavelengths.

measurements[7-9]. In a recent experiment, this technique was also used to evaluate the phase-noise spectral density of an optical atomic clock[10].

Here we use an optical atomic clock transition in a single-trapped $^{88}Sr^+$ ion and the quantum lock-in technique for the purpose of force metrology. Using the quantum lock-in method, we were able to measure force magnitude of $8.64 \pm 0.03 \times 10^{-19}$ N, at frequency of 1 kHz, three orders of magnitude below the trap resonance. The inferred sensitivity is $2.8 \times 10^{-20}$ N Hz$^{-1/2}$. We demonstrate two force detection methods for two scenarios—one in which the phase of the oscillating force is known and the quantum lock-in sequence can be synchronized with it, and one where only the force frequency is known.

## Results

**Experimental apparatus.** Our force detection is performed on an $^{88}Sr^+$ ion trapped in a linear Paul trap with radial frequencies close to 2.3 MHz and axial frequency of 1.13 MHz, and Doppler cooled to a temperature of $T = 2$ mK on a strong dipole allowed transition at 422 nm. The optical clock transition we use for force measurement is the $S_{1/2} \rightarrow D_{5/2}$ electric quadrupole transition at a wavelength of 674 nm and a natural linewidth of $\sim 0.4$ Hz. The clock laser beam crosses the trap axis, and therefore the direction of ion motion at an angle of 45°. At our temperature, the resulting Lamb-Dicke parameter for this transition, $\eta = 0.06$, is sufficiently small to allow for resolved motional sidebands to be observed and a Doppler-free carrier transition. Detection of successful electron-shelving to the $D_{5/2}$ level is performed through state-selective fluorescence on the 422 nm transition. The meta-stable $D$ levels are depleted using two infra-red re-pump lasers. A schematic diagram of the atomic levels involved is shown in Fig. 1. More details about our experimental set-up can be found in ref. 11.

Our clock laser is based on an external-cavity diode laser that is pre-stabilized to a high-finesse ($f = 100,000$) cavity made of ultra-low expansion glass. To filter out high-frequency phase noise, we use the cavity as a narrow optical transmission filter. The light transmitted through this cavity is injected into a bare slave diode laser[12]. This injected diode laser is further stabilized to another high-finesse ($f = 300,000$) ultra-low expansion glass cavity. The resulting laser linewidth at a time of 100 s is < 100 Hz. Slow drifts (0.1 Hz s$^{-1}$) in the cavity resonance frequency are mitigated by periodic calibration of the laser frequency to the optical clock transition. The two, optically separated, levels used for force detection are $(5S_{\frac{1}{2}}, m_1 = -\frac{1}{2})$ and $(4D_{\frac{5}{2}}, m_5 = \frac{1}{2})$, hereafter denoted as $|S, -\frac{1}{2}\rangle$ and $|D, \frac{1}{2}\rangle$, respectively.

**Force detection method description.** Under a constant-amplitude driving force that is far from resonance, the ion harmonic oscillator steady-state motion is given by,

$$x(t) = x_0 \cos(2\pi f_m t) = \frac{F_0}{4\pi^2 (f_t^2 - f_m^2) m_{ion}} \cos(2\pi f_m t). \quad (1)$$

Here, $F_0$ is the force amplitude, $m_{ion} = 87.9$ a.m.u. is the ion mass, $f_m = 1.013$ kHz and $f_t = 1.13$ MHz are the force driving frequency and trap harmonic frequency, respectively. The amplitude of the ion motion is linearly proportional to the amplitude of the driving force with a linear coefficient $(4\pi^2 (f_m^2 - f_t^2) m_{ion})^{-1}$. With all above parameters pre-determined, $F_0$ can be estimated from a measurement of the ion motion amplitude.

The amplitude of motion is determined by a measurement of the associated Doppler shifts of the clock transition using Ramsey spectroscopy. Initializing the ion in an equal superposition of the clock states, it will evolve in time as,

$$\psi(t) = \frac{1}{\sqrt{2}} \left( \left| S, -\frac{1}{2} \right\rangle + e^{i\phi_{clk}(t)} \left| D, \frac{1}{2} \right\rangle \right), \quad (2)$$

where $\phi_{clk}(t) = \int_0^t \delta(\tau') d\tau'$ and $\delta(\tau)$ is the frequency difference between the resonance frequency of the ion transition in its rest frame and the laser frequency at time $\tau$. The periodically driven ion motion results in a periodic Doppler shift and therefore,

$$\phi_{clk}(t) = \frac{2\pi x_0}{\sqrt{2}\lambda} [\sin(2\pi f_m t + \xi) - \sin(\xi)]. \quad (3)$$

Here, $\lambda$ is the laser wavelength and $\xi$ is the phase of the driving force at the beginning of the Ramsey sequence.

Since the clock superposition phase is periodically oscillating, for $\xi = 0$ it is maximal at quarter of the force cycle, but will have a zero average at longer times. Moreover, optical phase noise will lead to dephasing of the optical clock and loss of phase information. To accumulate the Doppler-shift phase over multiple force cycles and prolong the clock coherence, we used the quantum lock-in method[7]. To this end, a train of optical echo pulses modulated the clock superposition. Each block in the modulation sequence is composed of a wait time $\tau$ followed by a $\pi$-pulse on the optical clock transition and another equal wait time $\tau$. This block is repeated $n$ times. Using this sequence the superposition phase is modulated during the Ramsey sequence; $\phi_{clk}(t) = \int_0^t \delta(\tau', \xi) F_{mod}(\tau', \tau, n) d\tau'$; where the modulation function,

$$F_{mod}(t, \tau, n) = \Pi\left(\frac{t}{2n\tau}\right) \left[\Theta(t) + 2\sum_{k=1}^{\infty} (-1)^k \Theta(t - (2k-1)\tau)\right], \quad (4)$$

is a square-wave modulation, where $\Pi\left(\frac{t}{2\tau n}\right)$ is a rectangular function that nulls outside the range $|t| < \tau n$, where it is equal unity and $\Theta(t)$ is the Heaviside step function. An example modulation for $n = 4$ is plotted in Fig. 2. Modulating the phase improves on the measurement signal-to-noise ratio in two ways. First, by choosing $\tau$ and $\xi$ correctly, for example, $\tau = \frac{1}{4f_m}$ and $\xi = 0$, the phase of the superposition accumulates over multiple force cycles rather than oscillating around a zero mean. For the above choice, the accumulated phase measured at the end of the echo pulse sequence will be $\phi_{clk}(t = 2n\tau) = \frac{4\pi^2}{\sqrt{2}\lambda} f_m x_0 \int_0^{2n\tau} |\cos(2\pi f_m \tau')| d\tau' = n\frac{4\pi x_0}{\sqrt{2}\lambda}$. In addition, this modulation dynamically decouples the ion from noise components at frequencies other than harmonic multiples of $\frac{1}{4\tau}$ (ref. 13). It should be noted that other modulation sequences can be chosen as well, as long as they have a good spectral overlap with the force frequency. Here we use a train of echo pulses, which results-in a square-wave phase modulation, due to

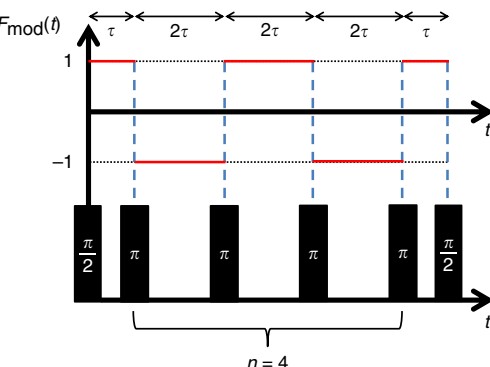

**Figure 2 | Schematic modulation function timeline.** Modulation function $F_{mod}(t, \tau, n = 4)$ (upper timeline) and the corresponding experimental pulse sequence (lower timeline).

its implementation simplicity and short exposure of the superposition states to light.

**Force measurement for known oscillatory force phase case.** The experiment sequence was as follows. The ion was constantly driven by an oscillating force resulting in periodic motion as in equation (1). The force was generated by modulating the voltage on one of the electrodes that create the trapping potential in the trap axis (endcap electrodes). The electronic state of the ion was initialized to $\left| S, -\frac{1}{2} \right\rangle$, using optical pumping to $\left| S, \frac{1}{2} \right\rangle$ followed by a radio-frequency $\pi$ pulse. After initialization, a clock laser $\frac{\pi}{2}$ pulse initialized the equal superposition of clock state in equation (2). This pulse was timed by an external trigger at a constant phase of the driving force to determine $\xi$. Subsequently, the modulation sequence of $n$ echo pulses described above, with $\tau = \frac{1}{4f_m}$, was executed. The sequence concluded with a second clock laser $\frac{\pi}{2}$ pulse, with laser phase $\phi$ relative to the initial $\frac{\pi}{2}$ pulse phase. Following the Ramsey experiment, the ion state was detected using state-selective fluorescence.

Figure 3a shows the probability of finding the ion in the $D$ level after the quantum lock-in sequence, as a function of the second $\frac{\pi}{2}$ pulse phase, $\phi$. The superposition phase, $\phi_{clk}$, was estimated by a maximum likelihood fit of our data to $f(\phi) = \frac{1}{2} + \frac{C}{2} \cos(\phi - \phi_{clk})$, where $C$ accounts for fringe contrast reduction due to dephasing. Such a reduction in contrast will compromise the phase estimation accuracy. To extract the modulating force from the $\phi_{clk}$, $\xi$ and $f_m$ have to be precisely known. To find the values of $\tau$ and $\xi$ that provide the maximum phase accumulation, we scanned these parameters and measured $\phi_{clk}$ for each $\tau$ and $\xi$ values. The expected $\phi_{clk}$ for every $\tau$ and $\xi$ can be analytically calculated to be,

$$\phi_{clk} = \frac{4\pi x_0}{\sqrt{2}\lambda} \frac{\cos\left(2\pi f_m n\tau + \xi + n\frac{\pi}{2}\right)}{\cos(2\pi f_m \tau)} \sin\left(2\pi f_m n\tau - n\frac{\pi}{2}\right) \sin^2(\pi f_m \tau). \quad (5)$$

The theoretically calculated $\phi_{clk}$ versus $\tau$ and $\xi$, and our experimental results are shown in Fig. 3b,c, respectively, for a lock-in sequence with $n = 10$ and $f_m = 1.013$ kHz. As seen, our results are in good agreement with the theoretical predictions. We estimated the motion amplitude of the ion to be $x_0 = 117.5 \pm 0.5$ nm, by a maximum likelihood fit of equation (5) to the data. In our measurement, laser phase noise as well as magnetic field noise can introduce superposition dephasing that might compromise the contrast of the Ramsey fringe and hence limit the precision of its phase estimation. Here, no significant reduction of contrast was observed, and hence we infer that the

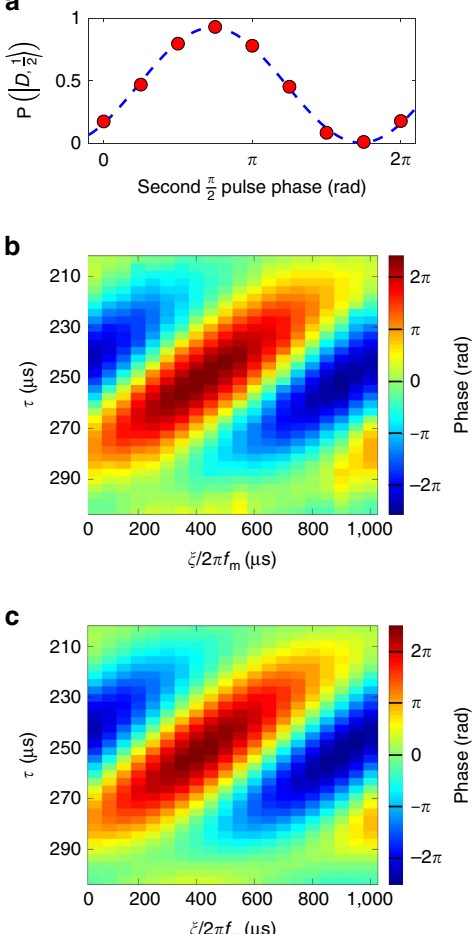

**Figure 3 | Experimental results of quantum lock-in force measurement.**
(**a**) A typical measured Ramsey fringe (red full circles) and fit to theory (dashed blue line) from which the superposition phase is estimated. (**b**) Measured phase as a function of $\tau$ and $\xi$. Each point on the plot corresponds to the phase extracted from a measurement similar to **a**. The phase offset $\xi$ is affected by different technical delays in our system. (**c**) Theoretical calculation for **b** using equation (5) with $A = \frac{2\pi x_0}{\sqrt{2}\lambda}$ and $\xi$ as fitted from **b**. The calculated value for the phase amplitude is $A = 0.774 \pm 0.004$ rad, corresponding to $x_0 = 117.5 \pm 0.5$ nm and $F_0 = 8.64 \pm 0.03 \times 10^{-19}$ N.

uncertainty in our measurement was dominated by quantum projection noise. Using equation (1), we inferred a force magnitude of $F_0 = 8.64 \pm 0.03 \times 10^{-19}$ N. With a total experiment time of roughly 9 h, our force estimation uncertainty is $3 \times 10^{-21}$ N, leading to sensitivity of $5.3 \times 10^{-19}$ N Hz$^{-1/2}$. Taking only the Ramsey fringe measurement corresponding to the optimal values of $\tau$ and $\xi$ into account, we achieve force uncertainty of $6 \times 10^{-21}$ N, for an experiment time of roughly 19 s. The sensitivity obtained from the single-fringe measurement is then $2.8 \times 10^{-20}$ N Hz$^{-1/2}$.

**Force measurement for unknown oscillatory force phase case.** Thus far we treated the case, in which the frequency and the phase of force exerted on the ion are coherently locked with respect to to the lock-in sequence. However, in many cases, the phase of the force is unknown, and perhaps also changes with time. In the case, where there is a continuous force spectrum, decoherence spectroscopy can be used in the limit of a Gaussian

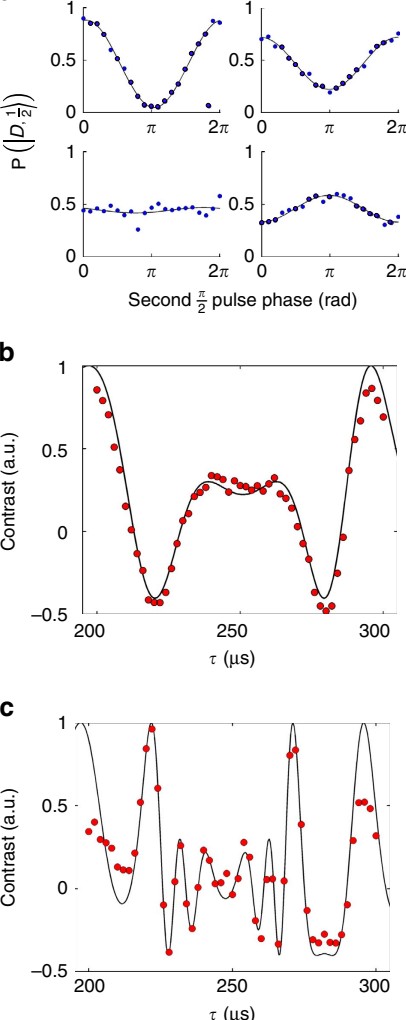

**Figure 4 | Experiment results of a-synchronous force measurement.**
(**a**) Four different contrast fringes measured (blue circles) and fitted to data (solid line) at different modulation frequencies for $n = 10$ echo pulses. As the ion modulation frequency approaches the force modulation frequency (1.013 kHz) the phase variance increases and the fringe contrast decays. The last fringe (1.136 kHz) shows partial re-phasing due to the topology of the Bloch sphere. (**b,c**) Measured contrast (red full circles) and a fit to theory (solid line) as a function of $\tau$ with $n = 10$, 20 echo pulses correspondingly. Each point on the plot corresponds to the contrast extracted from a measurement similar to **a**. The fit corresponds to equation (6). The calculated value for the ion motion amplitudes is $x_0 = 116 \pm 3$ nm and $x_0 = 115 \pm 4$ nm, and the corresponding force amplitudes are $F_0 = 8.54 \pm 0.2 \times 10^{-19}$ N and $F_0 = 8.47 \pm 0.3 \times 10^{-19}$ N for $n = 10$, 20 pulses, respectively.

distributed or sufficiently weak force that imparts small ($<< 2\pi$) phases on the clock superposition to estimate the force spectral density[14,15,16]. In cases, where the force spectrum is composed of discrete tones, the clock superposition can be used as force frequency and amplitude detector with sub-Fourier frequency estimation[13]. To demonstrate such incoherent force estimation, we repeated our measurements without triggering the quantum lock-in sequence at a constant force phase. Since the experiment is repeated at a rate that is incommensurate with the force frequency, the phase $\xi$ is sampled with uniformed probability in different repetitions of the experiment. After averaging over all possible $\xi$, the quantum lock-in fringe contrast can be shown to

be given by[7],

$$C(\tau, n) = \frac{1}{2} + \frac{1}{2} J_0 \left( \frac{4\pi x_0}{\sqrt{2}\lambda} \frac{\sin\left(2\pi f_{\mathrm{m}} n\tau - n\frac{\pi}{2}\right) \sin^2(\pi f_{\mathrm{m}}\tau)}{\cos(2\pi f_{\mathrm{m}}\tau)} \right), \quad (6)$$

where $J_0$ is the zero-order Bessel function of the first kind. The contrast, $C(\tau,n)$ is estimated by maximum likelihood fits to quantum lock-in phase scans as in the coherent force detection case. Four example fringes are shown in Fig. 4a showing the effect of noise with the same force amplitude, but different $\tau$ values. As seen, initially $C(\tau,n)$ decreases with increased $\tau$, resulting in larger phase variance due to the random Doppler shift, until the phase variance approaches $\frac{\pi}{2}$ and the fringe contrast nearly vanishes. However, as $\tau$ is further increased the fringe contrast re-appears with a $\pi$ phase shift (negative $C(\tau,n)$). This is due to the fact that when the phase variance approaches $\pi$, the different realizations partially re-phase and overlap. The measured contrast, $C(\tau,n)$, for an un-synchronized force detection is shown in Fig. 4b,c for $n = 10$, 20, respectively. The endcap modulation was similar to the synchronized experiment—amplitude of 30 mV and frequency of $f_{\mathrm{m}} = 1.013$ kHz. Each contrast measurement was performed twice—with and without a driving force. The measured contrast was taken to be the ratio between the two measurements, to reject contrast decrease due to unrelated effects, such as laser phase noise or $D$ level spontaneous decay. The solid line is a maximum likelihood fit of equation (6) to our data. We used two fit parameters, the motion amplitude $x_0$ and the modulation frequency $f_{\mathrm{m}}$. As seen, the theoretical prediction of equation (6) and our measured data are in good agreement. The frequency we measure is $f_{\mathrm{m}} = 1,015 \pm 2$ Hz. In both cases, the fitted value for $f_{\mathrm{m}}$ was within a 4 Hz wide 95% confidence interval around the actual value. As mentioned above, with this method the frequency of the force can be estimated with sub-Fourier uncertainty[7], that results from the non-linear response of the ion to the force drive that generates the narrow features appearing in Fig. 4b,c. The motion amplitude was estimated by the fit to be $x_0 = 116 \pm 3$ nm and $x_0 = 115 \pm 4$ nm, corresponding to $F_0 = 8.54 \pm 0.2 \times 10^{-19}$ N and $F_0 = 8.47 \pm 0.3 \times 10^{-19}$ N for $n = 10$, 20 pulses, respectively. The measurement sensitivity for $n = 10$, 20 was $1.7 \times 10^{-18}$ and $2 \times 10^{-18}$ NHz$^{-1/2}$, respectively. This result agrees with the force amplitude estimated in the case when the force phase was known.

## Discussion

The result reported above was obtained without reaching the sensitivity limit imposed by our system. The sensitivity reported here is shot-noise limited, and can be improved by more averaging. We did not perform an Allen deviation analysis as we aimed at a proof of principle demonstration, but we believe that measuring force amplitudes lower than Zepto-Newton range is possible using our method. The main limitation for measuring forces at frequencies $<1$ kHz in our system is laser phase noise, and hence with improved laser phase-noise force frequency $<1$ kHz can be detected.

To conclude, we demonstrated an oscillating force detection method using a single-trapped ion. Our method uses an optical clock superposition phase as a measure of the ion amplitude of motion. We used a quantum lock-in technique to amplify the desired signal, while reducing the effect of unwanted spectral components with dynamic decoupling. This technique allows us to measure forces at low frequencies, much lower than the trapping frequency. We demonstrated different methods for the case, in which the phase of the oscillating force is known and for the case when it is unknown, and we got sensitivity of $2.8 \times 10^{-20}$ NHz$^{-1/2}$. This sort of measurement can be useful for sensitive detection of electric fields close to a surface.

During the writing stages of this manuscript, we became aware of a similar theoretical proposal for force sensing[17].

**Data availability**. All relevant data presented in this paper are available from the corresponding author on request.

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

## Acknowledgements

This work was supported by the Crown Photonics Center, ICore-Israeli excellence center circle of light, the Israeli Science Foundation, the Israeli Ministry of Science Technology and Space, and the European Research Council (consolidator grant 616919-Ionology).

## Author contributions

R.S. and R.O. designed the experiment. R.S. performed the experiment and analysed the data. R.O. supervised the work. R.S. and R.O. wrote the manuscript together.

## Additional information

**Competing financial interests:** The authors declare no competing financial interests.

