## [Peer Review File · Nature Communications]

Reviewers' comments:

Reviewer #1 (Remarks to the Author):

This paper experimentally demonstrates a method for measuring oscillatory forces on a trapped single ion. The significant advance with this work is to combine a fairly standard methodology based on Doppler sensitivity with an established quantum lock-in technique to enable greater sensitivity and thus allow non-resonant driving forces to be measured as well as resonant ones. This possibility to measure non-resonant forces in ion traps is what makes this research novel and could lead to wider interest.

The paper is very well written, with clear structure and careful explanations throughout, enabling others to replicate these techniques if desired. The quality of experimental data is convincing and the conclusions appear to be reliable. I have just a few small comments on the text, which the authors may wish to address:

1) There appears to be an inconsistency of sign in the labelling of m_J states. At the bottom of page 4, ($5S_{\{1/2\}}$, $m_{\{1/2\}} = -1/2$) is denoted as $|S,+1/2\rangle$. Equation 2) uses $|S,+1/2\rangle$ as well, but page 6 tells us that the ion is initialised into $|S,-1/2\rangle$.

2) Since the 674nm transition is magnetically sensitive, I would expect magnetic field noise to be a further source of dephasing, in addition to the laser noise. Since magnetic field noise is never mentioned in the paper, perhaps it is significantly lower than the laser noise? Perhaps the authors might like to comment whether that is the case in their system, and whether they went to any special lengths to control magnetic fields to allow the quantum lock-in technique to be used over longer probe times?

3) Page 7, paragraph beginning "Figure 2a shows..." contains a statement, "To avoid systematic uncertainties in these two parameters, we scanned both τ and ξ ...". It is unclear to me how scanning avoids systematic uncertainties. For sure, it helps you see where ϕ_{clk} takes its maximum experimental value, but it doesn't ensure that this is free from all experimental biases.

4) The description of the experimental setup for the synchronous driving force never mentions how the oscillatory force is generated. This information comes only on page 9 (endcap modulation with 30mV amplitude). I think this information should come sooner.

5) Citations 8 and 9 are given as examples of state-of-the-art frequency comparisons. More advanced measurements have been made in

a) N. Huntemann et al, Phys Rev Lett 116, 063001 (2016)

b) T.L. Nicholson et al, Nature Comms 6, 6896 (2015)

Reviewer #2 (Remarks to the Author):

The manuscript presents experimental demonstration of oscillating-force detection method using single trapped ion. The external force oscillates with frequency much lower than the trap harmonic frequency, which induces a very small amplitude of motion. The authors use Doppler-shift measurement techniques to extract the amplitude of motion and thus to detect the amplitude of the external force.

In the Ramsey-type experiment, the ion is prepared in the equal superposition of the clock states. The state evolves in time, which leads to an additional phase in the superposition state. In order to

prolong the coherence time, the authors use the quantum lock-in method, in which a train of optical echo pulses modulated the clock superposition during the Ramsey sequence. The superposition phase is estimated by measuring the ion state using state-selective fluorescence. The authors report force sensitivity of $0.5 \text{ aN per } \sqrt{\text{Hz}}$.

The authors demonstrate also incoherent force estimation in which the phase of the force is unknown. The reported force sensitivity is of order of $3 \text{ aN per } \sqrt{\text{Hz}}$.

The results are interesting and the manuscript is essentially well written.

I have only some minor comments to :

1. The force sensitivity in the experiment is short-noise limited . I would like to see how the force sensitivity is scaled as a function of the difference between the force frequency and the trap harmonic frequency. What limits the force sensitivity to reach sensitivity beyond aN range?

2. In the experiment the accumulated phase is modulated due to the train of n echo pulses. How the force sensitivity is improved by using higher n?

In summary, I think that the paper contains enough interesting results to be published in Nature Communications, and that it fits the scope of the journal.

Reviewer #3 (Remarks to the Author):

In this paper, Shaniv and Ozeri report on force-sensing using a trapped singly-charged 88Sr ion. They implement a lock-in technique based on dynamical decoupling which allows the motion of the trapped ion to be detected by its resulting Doppler shift, which changes the internal state of the atom following the experimental sequence. This technique allows them to measure a force that they apply using electric fields, at a frequency of around 1 kHz, which lies far below the ion trapping resonance frequency. The sensitivity achieved is impressive ($\text{few } 10^{-19} \text{ N}/\sqrt{\text{Hz}}$) e.g. when compared with force microscopy methods. Although in terms of raw sensitivity it falls short of prior work on resonant force sensing with trapped ions (e.g. Ref. 5), in the current work the authors have sought to detect lower-frequency forces off-resonance, which is significantly more challenging. I find that this is a useful application of the Quantum Lock-in sequence and will be of interest to the force sensing community. The writing and presentation of the data are clear and the overall quality of the work seems comparable with other manuscripts published in Nature Communications. My only quibbles with the manuscript are relatively minor, and are listed below. I can recommend publication after the 1st of these has been adequately addressed:

1) As discussed at the top of pg 8, the authors claim to measure an applied force with magnitude of $8.64 \times 10^{-19} \text{ N}$ for the case where the frequency and phase of the applied force is controlled with respect to the lock-in sequence. In the following sentence, they infer a force sensitivity of $5.3 \times 10^{-19} \text{ N}/\sqrt{\text{Hz}}$ but indicate a total experiment time of roughly 9 Hours. If this is the true sensitivity, even at 1 Hz bandwidth the authors could detect the applied force of $8.64 \times 10^{-19} \text{ N}$. Why is averaging for 9 hours required? This must be more clearly explained prior to publication. Otherwise I do not understand what is meant here by "force estimation sensitivity".

2) (minor) Several typographical errors appear throughout the manuscript, for example:

- At the bottom of page 2, "mean" should be "means".
- On page 3, 1st full paragraph, 2nd sentence "however" should be "However".
- On page 4, end of 1st paragraph, "15" should be "Ref. 15" or similar.

Below please find our detailed response to the referee's comments:

Reviewer #1:

- 1) "There appears to be an inconsistency of sign in the labelling of mJ states. At the bottom of page 4, ($5S_{-1/2}, m_{-1/2} = -1/2$) is denoted as $|S,+1/2\rangle$. Equation 2) uses $|S,+1/2\rangle$ as well, but page 6 tells us that the ion is initialized into $|S,-1/2\rangle$."

Our reply:

We thank the referee for pointing to this typo. In the experiment we initialized our ion to the $\left|5S_{\frac{1}{2}}, m_{\frac{1}{2}} = -\frac{1}{2}\right\rangle := \left|S, -\frac{1}{2}\right\rangle$ state. We corrected the bottom of page 4 and equation 2.

- 2) "Since the 674nm transition is magnetically sensitive, I would expect magnetic field noise to be a further source of dephasing, in addition to the laser noise. Since magnetic field noise is never mentioned in the paper, perhaps it is significantly lower than the laser noise? Perhaps the authors might like to comment whether that is the case in their system, and whether they went to any special lengths to control magnetic fields to allow the quantum lock-in technique to be used over longer probe times?"

Our reply:

We agree with the referee that both laser phase noise and magnetic field noise introduce the same limitations to our measurement scheme through dephasing. We do not know which of these two is the main source of dephasing at exactly the modulation frequencies used here. Magnetic field noise was not mitigated by working with line-trigger and no active or passive noise cancelation was used. By comparing simple Ramsey and single echo experiments between an S-> D superposition to spin superpositions in the ground state - we found that, at least close to dc, our S->D coherence is mainly limited by laser phase noise. However, as mentioned above, no such analysis was performed at the 1 kHz modulation frequency used in our force detection. We added a mention of the possibility of magnetic field contributions to dephasing.

- 3) "Page 7, paragraph beginning "Figure 2a shows..." contains a statement, "To avoid systematic uncertainties in these two parameters, we scanned both tau and xi... ". It is unclear to me how scanning avoids systematic uncertainties. For sure, it helps you see where phi_clk takes its maximum experimental value, but it doesn't ensure that this is free from all experimental biases."

Our reply:

We agree with the referee that our 2D scan does not eliminate most systematic biases. We therefore corrected the sentence in page 7 to "To find the values of τ and ξ that provide the maximum phase accumulation, we scanned these parameters and measured ϕ_{clk} for each τ and ξ values."

- 4) "The description of the experimental setup for the synchronous driving force never mentions how the oscillatory force is generated. This information comes only on page 9 (endcap modulation with 30mV amplitude). I think this information should come sooner. "

Our reply:

As the reviewer suggested, we now elaborate on the force generation method earlier in the manuscript, in the same section where the description of the experiment appears (page 7, top).

- 5) "Citations 8 and 9 are given as examples of state-of-the-art frequency comparisons. More advanced measurements have been made in
a) N. Huntemann et al, Phys Rev Lett 116, 063001 (2016)
b) T.L. Nicholson et al, Nature Comms 6, 6896 (2015)"

Our reply:

We thank the referee for pointing out these references and have included them as well.

Reviewer #2:

1. "The force sensitivity in the experiment is short-noise limited . I would like to see how the force sensitivity is scaled as a function of the difference between the force frequency and the trap harmonic frequency. What limits the force sensitivity to reach sensitivity beyond aN range? "

Our reply:

As the referee points out, we did not push the limits of our measurement sensitivity to those imposed by drifts in system parameters or other types of noise. In the absence of noise the motional amplitude depends on the difference between the trap and drive frequencies as described by Eq. (1). The accumulated phase and therefore the measurement sensitivity are linear in this amplitude (Eq. 5). In our system the main limitation for measuring forces at lower frequencies than 1 kHz is laser phase noise. Force sensitivity below the aN range is perhaps possible with more averaging. We did not perform a long time Allen deviation analysis to find where the limits are - as we aimed at a proof of principle demonstration of this technique. Since the limits are determined by the specific noise characters of each system this characterization is system specific and not generic to the method. We suspect that in our system, force sensitivity below aN range would eventually be limited by the stability of the voltage modulation we apply on the trap electrodes to generate force modulation. In fact, since our laser fast linewidth is around 100 Hz at 100 sec, optical clock labs with laser linewidths below 1 Hz would be able to measure much lower frequency forces with better sensitivity. We elaborate on the prospects of such low frequency force measurements in the introduction.

2. "In the experiment the accumulated phase is modulated due to the train of n echo pulses. How the force sensitivity is improved by using higher n"

Our reply:

The dependence of the accumulated phase, and therefore also the sensitivity to force is linear in the number of applied pulses, n,

$$\phi_{clk}(t = 2n\tau) = \frac{4\pi^2}{\sqrt{2\lambda}} f_m x_0 \int_0^{2n\tau} |\cos(2\pi f_m \tau')| d\tau' = n \frac{4\pi x_0}{\sqrt{2\lambda}}$$

This is because the same phase is acquired in every half cycle of motion. We added this equation to the discussion below Eq. 4.

Reviewer #3:

1. "As discussed at the top of pg 8, the authors claim to measure an applied force with magnitude of 8.64×10^{-19} N for the case where the frequency and phase of the applied force is controlled with respect to the lock-in sequence. In the following sentence, they infer a force sensitivity of 5.3×10^{-19} N/ $\sqrt{\text{Hz}}$ but indicate a total experiment time of roughly 9 Hours. If this is the true sensitivity, even at 1 Hz bandwidth the authors could detect the applied force of 8.64×10^{-19} N. Why is averaging for 9 hours required? This must be more clearly explained prior to publication. Otherwise I do not understand what is meant here by "force estimation sensitivity". "

Our reply:

The referee is correct in stating that the force of 8.64×10^{-19} N could have been detected in our system after only 1 sec averaging. In fact, the force magnitude we applied was arbitrary and we averaged for 9 hours in order to see how much we can reduce the **uncertainty** with which this force magnitude is estimated. After 9 hours the uncertainty was reduced to 3×10^{-21} N which is the important number we report, rather than the magnitude of force. We modified the manuscript to emphasize this point. Furthermore, the strategy with which we estimated this force was far from optimal. As can be seen in figure 2, many of these 9 hours were spent on ξ and τ values which contain no information on the force amplitude. With measuring Ramsey fringes that are taken at the optimal $\xi = 0$ and $f_m \tau = \frac{1}{4}$ we can estimate the force with much better sensitivity of $2.8 \times 10^{-20} \frac{\text{N}}{\sqrt{\text{Hz}}}$. We modified the manuscript to reflect this point and we changed the sensitivity reported in the abstract accordingly.

2. "Several typographical errors appear throughout the manuscript, for example:
 - At the bottom of page 2, "mean" should be "means".
 - On page 3, 1st full paragraph, 2nd sentence "however" should be "However".
 - On page 4, end of 1st paragraph, "15" should be "Ref. 15" or similar. "

Our reply:

We thank the referee for pointing these typos to us. Corrected.

REVIEWERS' COMMENTS:

Reviewer #1 (Remarks to the Author):

I am satisfied that the authors have addressed all my comments now, and I consider the paper ready for publication in Nature Communications.

Reviewer #2 (Remarks to the Author):

This reviewer provided confidential remarks to the editor with a recommendation to publish the manuscript in Nature Communications.

Reviewer #3 (Remarks to the Author):

After considering the response to the comments from all of the reviewers, I believe the manuscript has been improved. In particular I find the authors reply to the 2nd reviewer's question regarding the limitations of the force sensitivity to be illuminating. I was disappointed however that more of this response was not captured in the revised manuscript, as I think this is a question that will be common for many readers. In particular it is very useful for the community to understand the limitations of the technique. I suggest the authors expand the discussion of this in the manuscript to reflect the content in the reply, e.g. what limits the force sensitivity to reach beyond aN range, including phase noise, noise/stability of the voltage modulation, magnetic field noise etc. Otherwise I am satisfied with the revised manuscript and recommend proceeding with publication.

Reviewer #3 comments:

After considering the response to the comments from all of the reviewers, I believe the manuscript has been improved. In particular I find the authors reply to the 2nd reviewer's question regarding the limitations of the force sensitivity to be illuminating. I was disappointed however that more of this response was not captured in the revised manuscript, as I think this is a question that will be common for many readers. In particular it is very useful for the community to understand the limitations of the technique. I suggest the authors expand the discussion of this in the manuscript to reflect the content in the reply, e.g. what limits the force sensitivity to reach beyond aN range, including phase noise, noise/stability of the voltage modulation, magnetic field noise etc. Otherwise I am satisfied with the revised manuscript and recommend proceeding with publication.

Our reply:

We thank the reviewer for pointing out an issue that can be interesting to the readers. We added a paragraph in our Discussion section mentioning possible limitations to our force measurement method:

"The result reported above was obtained without reaching the sensitivity limit imposed by our system. The sensitivity reported here is shot-noise limited, and can be improved by more averaging. We did not perform an Allen deviation analysis as we aimed as a proof of principle 10 demonstration, but we believe that measuring force amplitudes lower than aN range is possible using our method. The main limitation for measuring forces at frequencies lower than 1 kHz in our system is laser phase noise, and hence with improved laser phase noise force frequency lower than 1 kHz can be detected".